# Effects of Vitamin E on Redox balance in regulating thiol/disulfide homeostasis in sepsis: An antioxidant therapy perspective

Veli Fahri Pehlivan[1]*, Başak Pehlivan[1], Erdogan Duran[1], Abdullah Taskın[2], Ismail Koyuncu[3], Yusuf Çakmak[4]

1 Faculty of Medicine, Department of Anesthesia and Reanimation, Harran University, Sanlıurfa, TÜRKİYE,
2 Harran University, Health Science Faculty, Department of Nutrition and Dietetics, Sanlıurfa, TÜRKİYE,
3 Faculty of Medicine, Department of Medicinal Biochemistry, Harran University, Sanlıurfa, TÜRKİYE,
4 Harran University, Animal Experiment Application and Research Center (HDAM), Sanlıurfa, TÜRKİYE

* vfpehlivan@harran.edu.tr, vfpehlivan@gmail.com

## Abstract

### Background

Sepsis, a life-threatening condition resulting from a dysregulated host response to infection, is associated with high mortality and remains a major global health burden. Sepsis is characterized by an imbalance between oxidative stress and inflammation, leading to disruption of thiol–disulfide homeostasis, hematological abnormalities, cytokine dysregulation, and widespread tissue injury.

### Methods

An experimental sepsis model was established in thirty-two male Balb-C mice using lipopolysaccharide administration. Animals were randomized into four groups: control, vitamin E, sepsis, and sepsis plus vitamin E. Serum oxidative stress markers, thiol-disulfide parameters, and inflammatory mediators, including C-reactive protein, interleukin-40, and tumor necrosis factor-alpha, were measured. Hematological indices of systemic inflammation were evaluated (Neutrophil-to-Lymphocyte Ratio, Platelet-to-Lymphocyte Ratio), and lung, liver, and kidney tissues were examined histologically using a semi-quantitative scoring system.

### Results

Lipopolysaccharide-induced sepsis caused marked disruption of thiol-disulfide balance, characterized by reduced native and total thiol levels, elevated disulfide levels, increased cytokine release, and severe histopathological injury. Vitamin E supplementation restored thiol-disulfide homeostasis, decreased oxidative stress, and attenuated systemic inflammation. In the sepsis plus vitamin E group, serum thiol levels increased significantly, while disulfide levels declined. Interleukin-40 showed

**Data availability statement:** All relevant data are within the manuscript and its Supporting Information files.

**Funding:** The author(s) received no specific funding for this work.

**Competing interests:** The authors have declared that no competing interests exist.

a 24.2% reduction and tumor necrosis factor-alpha a 9.8% reduction compared with untreated septic animals. Histopathological analyses confirmed reduced inflammatory cell infiltration, vascular congestion, and tissue degeneration, particularly in the lungs.

## Conclusions

Vitamin E demonstrated significant protective effects against sepsis-induced oxidative and inflammatory injury by preserving thiol-disulfide homeostasis and reducing cytokine production. The more pronounced effect on interleukin-40 compared with tumor necrosis factor-alpha suggests selective modulation of inflammatory pathways and highlights interleukin-40 as a potential biomarker and therapeutic target. These findings support vitamin E as a promising adjunctive therapy in sepsis, although further studies are required to define optimal dosing strategies and assess clinical applicability.

## Introduction

Sepsis is a critical medical condition characterized by a systemic inflammatory response to infection, leading to widespread organ dysfunction and high mortality rates [1,2]. The pathophysiology of sepsis involves a complex cascade of events, including uncontrolled inflammation, oxidative stress, and immune dysregulation [3,4]. Oxidative stress, defined as an imbalance between the production of reactive oxygen species (ROS) and the body's ability to detoxify them, plays a pivotal role in the progression of sepsis by causing cellular damage and exacerbating the inflammatory response [3,5,6].

Thiol-disulfide homeostasis, a crucial redox balance system, is significantly disrupted during sepsis. Thiols, particularly protein thiols, are vital for maintaining cellular integrity and function, acting as potent antioxidants. The oxidation of thiols to disulfides is a reversible process, and an imbalance in this system reflects increased oxidative stress and impaired antioxidant defense mechanisms [3,7].

Interleukin-40 (IL-40), a newly identified cytokine secreted by activated B cells, has emerged as an important mediator in sepsis pathophysiology. Recent studies have demonstrated that IL-40 modulates sepsis-induced inflammation by blocking NETosis (Neutrophil Extracellular Trap formation) and may serve as both a prognostic biomarker and therapeutic target [8,9]. The ability of IL-40 to prevent multi-organ damage during sepsis highlights its potential significance in understanding and treating this complex condition.

Tumor necrosis factor-alpha (TNF-α), a central mediator of sepsis produced by macrophages and other immune cells, induces endothelial dysfunction, coagulopathy, and organ failure through inflammatory cascades [10,11]. TNF-α can reproduce all clinical signs of sepsis when administered systemically, underscoring its pivotal role in sepsis pathogenesis. Notably, vitamin E, particularly δ-tocotrienol, exerts anti-inflammatory effects by inhibiting TNF-α-induced NF-κB activation and enhancing anti-inflammatory A20 protein expression [12]. These insights emphasize the

complex interplay of oxidative stress, thiol-disulfide homeostasis, and key inflammatory mediators such as IL-40 and TNF-α in sepsis pathogenesis.

In clinical practice, hemogram parameters are essential for assessing sepsis severity and the inflammatory response [13,14]. Recently, derived ratios such as neutrophil-to-lymphocyte ratio (NLR), platelet-to-lymphocyte ratio (PLR), and neutrophil/lymphocyte × platelet ratio (NLPR) have gained importance as prognostic markers [15,16]. These indices are also thought to reflect oxidative stress through their association with thiol-disulfide homeostasis. Disruptions in both redox balance and hemogram-derived markers suggest a link between oxidative stress and inflammation in sepsis [7,14,17,18].

Vitamin E is a family of fat-soluble compounds consisting of tocopherols and tocotrienols (α, β, γ, δ), all of which possess strong antioxidant properties. Although α-tocopherol is the most common and biologically active form, recent studies have demonstrated that δ-tocotrienol may exert stronger effects, particularly in inflammatory processes [12]. Vitamin E mitigates oxidative stress by scavenging free radicals, protecting cell membranes against lipid peroxidation, and supporting the regeneration of other antioxidants such as vitamin C [11,19,20]. Moreover, its immunomodulatory effects are associated with the regulation of cytokine production, suppression of the NF-κB signaling pathway, and improvement of T-cell functions [12]. Collectively, these properties render vitamin E a potential therapeutic agent in clinical conditions characterized by pronounced oxidative stress and inflammation, such as sepsis.

The primary aim of this study is to comprehensively investigate the effects of vitamin E on redox balance, specifically focusing on thiol-disulfide homeostasis, and its impact on inflammatory markers including IL-40 and TNF-α, as well as hemogram parameters in an experimental LPS-induced sepsis model. This research seeks to clarify whether vitamin E administration can effectively restore redox balance and attenuate the systemic inflammatory response in sepsis.

This study advances current knowledge by providing detailed insights into the specific mechanisms through which vitamin E exerts its protective effects in sepsis, particularly concerning its influence on thiol-disulfide homeostasis and novel inflammatory mediators. While previous studies have explored the general antioxidant properties of vitamin E, our work uniquely highlights its direct role in maintaining critical redox balance during sepsis and its differential effects on emerging biomarkers like IL-40 compared to classical inflammatory mediators like TNF-α. Furthermore, by correlating these biochemical changes with alterations in hemogram parameters and histopathological findings, we offer a more holistic understanding of vitamin E's therapeutic potential in sepsis management.

## Methods

### Study design and animals

This experimental study was conducted using an LPS-induced sepsis model in male Balb-C mice. A total of thirty-two male Balb-C mice, weighing 25–30 g and aged 8–10 weeks, were obtained from the Experimental Animal Laboratory of Harran University (HDAM, Sanliurfa, Turkey). The animals were housed under controlled environmental conditions (22 ± 2°C, 12-hour light/dark cycle) with free access to standard rodent chow and water. All experimental procedures were approved by the Harran University Animal Experiments Ethics Committee (Approval No: 2024/006/003) and conducted in accordance with the National Institutes of Health Guide for the Care and Use of Laboratory Animals. Efforts were made to minimize animal suffering and reduce the number of animals used.

### Sample size determination

The sample size was determined using G*Power 3.1.9.7 software based on effect sizes reported in previous experimental studies (Cohen's d = 1.2). With α = 0.05 and β = 0.20 (power = 0.80), a minimum of six animals per group was required [21]. To account for potential attrition, eight animals per group (n = 8) were included. This sample size was considered sufficient to detect significant differences in key biochemical and inflammatory parameters, consistent with both preliminary experimental data and published literature, while adhering to ethical guidelines for animal research.

## Experimental groups and treatment protocol

Mice were randomly divided into four experimental groups (n = 8 per group):

1. **Control Group (DMSO):** Mice in this group received an intraperitoneal (i.p.) injection of Dimethyl Sulfoxide (DMSO), which served as the vehicle for vitamin E.

2. **Sepsis Group (LPS):** Mice in this group received a single i.p. injection of Lipopolysaccharide (LPS) (Escherichia coli O111:B4, Sigma-Aldrich, USA) at a dose of 10 mg/kg to induce sepsis.

3. **Vitamin E Group (Vit E):** Mice in this group received a single i.p. injection of vitamin E (alpha-tocopherol, Sigma-Aldrich, USA) at a dose of 100 mg/kg.

4. **LPS + Vit E Group**: Mice in this group received an i.p. injection of vitamin E (100 mg/kg) 30 minutes prior to the LPS injection (10 mg/kg). This protocol was designed to investigate the prophylactic/preconditioning effects of vitamin E against LPS-induced sepsis.

The vitamin E dose (100 mg/kg) was selected based on previous experimental studies demonstrating antioxidant efficacy within non-toxic ranges [22]. This dose corresponds to a human equivalent of approximately 8 mg/kg, which is considered clinically applicable. The timing of administration (30 minutes prior to LPS) was determined to allow sufficient time for cellular uptake of vitamin E and activation of antioxidant defense mechanisms.

All injections were administered in a total volume of 200 µL. Animals were monitored closely for 24 hours post-LPS injection for signs of sepsis and mortality. At the end of the experimental period, animals were humanely euthanized under deep anesthesia, and blood samples were collected via cardiac puncture for biochemical and hematological analyses. Tissue samples (e.g., liver, kidney, lung) were also collected for histopathological examination.

## Biochemical analysis

Serum thiol-disulfide homeostasis parameters, including native thiol, total thiol, and disulfide levels, were evaluated using the automated colorimetric method described by Erel and Neselioglu [3]. This method is based on the reduction of disulfide bonds to thiol groups, allowing for the spectrophotometric measurement of both native and total thiols, from which disulfide levels can be calculated. Results were expressed in µmol/L.

Serum Interleukin-40 (IL-40) and tumor necrosis factor-alpha (TNF-α) levels were quantified using commercial ELISA kits (Sunlong Medical, Cat No: SL1923Ra for IL-40 and EL0013Ra for TNF-α) according to the manufacturer's instructions. Briefly, samples were applied to 96-well microplates pre-coated with rat-specific antibodies, followed by incubation with biotinylated antibodies and streptavidin–HRP. After washing, substrate solution was added, and the reaction was stopped with an acidic solution. Absorbance was measured at 450 nm using a microplate reader (Cytation-1, BioTek). The assay sensitivities were 1.0 pg/mL for IL-40 and 5.0 pg/mL for TNF-α. All samples were analyzed in duplicate, and only results with a coefficient of variation below 10% were accepted. Calibration curves were generated for each assay, with $R^2$ values consistently above 0.99.

## Hematological analysis

Hemogram parameters, including white blood cell (WBC) count, neutrophil count, lymphocyte count, platelet count, and red blood cell (RBC) count, were measured using the Alinity HQ (Abbott, USA), a next-generation fully automated hematology analyzer. From these parameters, the Neutrophil-to-Lymphocyte Ratio (NLR), Platelet-to-Lymphocyte Ratio (PLR), and Neutrophil/Lymphocyte × Platelet Ratio (NLPR) were calculated as indicators of systemic inflammation. C-reactive protein (CRP) levels were measured using an enzyme-linked immunosorbent assay (ELISA) kit (R&D Systems, USA) according to the manufacturer's instructions.

## Histopathological examination

Formalin fixed, paraffin embedded lung, liver, and kidney tissue samples were sectioned at 5 μm thickness and stained with hematoxylin and eosin (H&E) for routine histopathological examination. Microscopic evaluation was performed by a blinded, board certified pathologist using a Nikon Eclipse light microscope at ×400 magnification, and representative images were captured with a Nikon DS-Fi2 camera (Nikon, Japan).

Histopathological changes were evaluated using a **semi-quantitative scoring system** based on the severity and extent of tissue injury: **0 = none, 1 = mild, 2 = marked, 3 = diffuse**. For lung tissue, parameters included alveolar damage, bronchial epithelial degeneration/desquamation, interstitial inflammation, emphysematous changes, and vascular congestion [23]. For liver tissue, hepatocyte vacuolar degeneration, portal inflammation, vascular congestion, and necrosis were assessed [24]. For kidney tissue, tubular degeneration/necrosis, glomerular changes, interstitial inflammation, and vascular congestion were scored [25].

## Statistical analysis

Statistical analyses were performed using SPSS software (Version 26.0, IBM Corp., Armonk, NY, USA). All continuous variables, including histopathological scores, are presented as **mean ± standard deviation (SD)**. The normality of data distribution was assessed using the **Shapiro–Wilk test**. For normally distributed data, differences among the four experimental groups (DMSO, Vitamin E, LPS, and LPS + Vitamin E) were analyzed using **one-way analysis of variance (One-Way ANOVA)**. When overall differences were statistically significant, **Tukey's post-hoc test** was applied for general biochemical and hematological parameters, while **Bonferroni-adjusted post-hoc multiple comparisons** were specifically used for histopathological scores to control for type I error.

For non-normally distributed data, the **Kruskal–Wallis test** was employed, followed by **Dunn's post-hoc test** for pairwise comparisons. Pearson's correlation coefficient was calculated to assess relationships between continuous variables. For histopathological evaluation, semi-quantitative scores (0 = none, 1 = mild, 2 = marked, 3 = diffuse) were assigned based on tissue-specific injury parameters, and groups sharing the same superscript letter were interpreted as not significantly different, whereas groups with different superscript letters were considered significantly different. A **p-value < 0.05** was accepted as the threshold for statistical significance. The minimal dataset containing all biochemical, hematological, cytokine, and histopathological parameters used in this study is provided as Supporting Information (Minimal_dataset_thiol_disulfide_sepsis.xlsx).

## Results

### Vitamin E Preserves Redox Homeostasis and Attenuates Inflammatory Response in Experimental Sepsis

Vitamin E supplementation significantly modulated both redox balance and inflammatory response in the experimental sepsis model. As demonstrated in Table 1, LPS administration led to a marked disruption of thiol-disulfide homeostasis, characterized by decreased native and total thiol levels alongside increased disulfide levels compared with the Control group (p < 0.001). In contrast, treatment with Vitamin E, either alone or in combination with LPS, preserved thiol-disulfide equilibrium, resulting in significantly higher native and total thiol levels and lower disulfide levels than in the Sepsis group (p < 0.001). These findings highlight the antioxidant role of Vitamin E in maintaining redox balance under septic conditions.

In parallel, inflammatory markers exhibited significant group differences. Serum levels of IL-40, a newly identified biomarker, and TNF-α, a classical proinflammatory cytokine, were highest in the LPS group, corroborating the successful establishment of the sepsis model and paralleling CRP elevation (p < 0.001). Both IL-40 and TNF-α levels were significantly lower in the DMSO and Vit E groups, and Vitamin E supplementation in the LPS + Vit E group exerted a significant suppressive effect compared with the LPS group (p < 0.05). Interestingly, this anti-inflammatory effect was more pronounced for IL-40 than for TNF-α, with IL-40 levels showing a 24.2% decrease compared to a 9.8% decrease in TNF-α levels, suggesting a selective modulatory action of Vitamin E on specific inflammatory pathways.

**Table 1. Effects of Vitamin E on Serum Thiol–Disulfide Homeostasis Parameters in Experimental Sepsis.**

| | DMSO | Vit E | LPS | LPS+Vit E | p value |
|---|---|---|---|---|---|
| **Native thiol, µmol/L** | 388.0±26.3 [α,β] | 509.7±64.0 [δ,ε] | 251.5±41.9 [Ω] | 440.6±32.5 | <0.001 |
| **Total thiol, µmol/L** | 448.5±27.6 [α,β] | 559.6±61.8 [δ,ε] | 332.6±42.9 [Ω] | 468.7±34.0 | <0.001 |
| **Disulfide, µmol/L** | 30.2±5.7 [β,γ] | 24.9±3.7 [δ,ε] | 40.5±5.8 [Ω] | 14.0±3.2 | <0.001 |
| **Oxidized thiol, %** | 6.74±1.24 [α,β,γ] | 4.51±0.92 [δ] | 12.33±2.09 [Ω] | 3.00±0.69 | <0.001 |
| **Reduced thiol, %** | 86.5±2.4 [α,β,γ] | 90.9±1.8 [δ] | 75.3±4.1 [Ω] | 93.9±1.3 | <0.001 |
| **Thiol Ox-Red, %** | 7.84±1.68 [β,γ] | 4.98±1.12 [δ] | 16.54±3.71 [Ω] | 3.20±0.77 | <0.001 |
| **CRP, mg/dL** | 0.58±0.04 [β] | 0.54±0.05 [δ] | 0.91±0.04 [Ω] | 0.58±0.03 | <0.001 |
| **IL-40, pg/mL** | 74.19±3.30 [α,β] | 66.34±4.57 [δ,ε] | 105.85±8.28 [Ω] | 80.20±5.57 | <0.001 |
| **TNF-α, pg/mL** | 177.28±23.46 [β] | 165.54±6.30 [δ,ε] | 223.08±37.33 | 201.23±21.29 | <0.001 |

(Data are expressed as mean ±standart deviation. α: DMSO vs Vit E, β: DMSO vs LPS, γ: DMSO vs LPS+Vit E, δ: Vit E vs LPS, ε: Vit E vs LPS+Vit E, Ω: LPS vs LPS+Vit E.)

Together, these results indicate that Vitamin E supplementation not only preserves thiol-disulfide homeostasis but also attenuates the inflammatory response in sepsis, underscoring its potential as an adjunctive therapeutic agent. However, the differential effects on IL-40 and TNF-α suggest that further investigations into dose optimization, timing, and mechanistic pathways are warranted to maximize clinical efficacy.

## Alterations in Hematological Indices and Inflammatory Biomarkers

Analysis of hemogram parameters and inflammatory markers revealed significant alterations in response to LPS-induced sepsis and subsequent vitamin E treatment (Table 2). The Sepsis group exhibited significantly increased NLR, PLR, and CRP levels compared to the Control group (p<0.05). In the Vit E and LPS+Vit E groups, vitamin E administration led to a significant reduction in CRP levels compared to the Sepsis group (p<0.05), indicating a clear anti-inflammatory effect. While NLR and PLR showed a trend towards normalization in the vitamin E-treated groups, these changes were not always statistically significant when compared directly to the Sepsis group, suggesting that while vitamin E modulates the inflammatory response, its impact on these specific ratios might be less pronounced or require higher doses/longer treatment durations to achieve statistical significance.

**Table 2. Hematological Parameters of Experimental Groups (WBC, NEU, LYM, PLT, MPV, NLR, PLR, NLPR).**

| | DMSO | Vit E | LPS | LPS+Vit E | p value |
|---|---|---|---|---|---|
| **WBC (10³/uL)** | 4.1 [2.0] | 5.7 [1.9] [ε] | 5.0 [0.5] | 3.5 [0.6] | 0.013 |
| **Neutrophil, 10³/uL** | 0.009 [0.01] | 0.034 [0.02] | 0.010 [0.03] | 0.016 [0.02] | 0.776 |
| **Lymphocyte, 10³/uL** | 0.49 [3.0] [α] | 3.37 [1.0] | 0.63 [2.1] | 2.48 [0.8] | 0.024 |
| **Platelets, 10³/uL** | 555 [193] | 910 [236] [δ] | 516 [106] [Ω] | 815 [227] | 0.009 |
| **MPV, fL** | 4.9 [0.2] [α] | 4.5 [0.3] | 4.7 [0.7] | 4.7 [0.3] | 0.037 |
| **NLR** | 0.008 [0.03] | 0.009 [0.01] | 0.006 [0.05] | 0.006 [0.01] | 0.885 |
| **PLR** | 1098 [1250] | 253 [211] | 841 [1081] | 319 [133] | 0.058 |
| **NLPR** | 0.001 [0.01] | 0.001 [0.00] | 0.001 [0.01] | 0.001 [0.00] | 0.776 |

Data are expressed as Median [IQR]. The Kruskal-Wallis test was used, and p-values of less than 0.05 were regarded as statistically significant. (WBC: White blood count; MPV: Mean Platelet Volume NLR: Neutrophil lymphocyte ratio; PLR: Platelet lymphocyte ratio; NLPR: Neutrophil/ lymphocyte*platelet ratio; α: DMSO vs Vit E; ε: Vit E vs LPS+Vit E; δ: Vit E vs LPS; Ω: LPS vs LPS+Vit E)

## Correlation Between Antioxidant Activity and Inflammatory Markers: Effect of Vit E

Table 3 demonstrates strong correlations between redox balance markers and inflammatory mediators, highlighting the intricate interplay between oxidative stress and inflammation in sepsis. Native thiol and total thiol levels were strongly and negatively correlated with CRP (r= −0.808 and r= −0.766, respectively; p<0.001), IL-40 (r= −0.818 and r= −0.811, respectively; p<0.001), and TNF-α (r= −0.601 and r= −0.642, respectively; p<0.001), reflecting the decline in antioxidant capacity under inflammatory stress. Conversely, disulfide levels correlated positively with CRP (r=0.660, p<0.001) and IL-40 (r=0.518, p=0.002), indicating oxidative stress–driven shifts in thiol–disulfide homeostasis. Importantly, the robust negative correlation between native thiol and IL-40 (r= −0.818, p<0.001) underscores a direct link between loss of antioxidant defenses and elevation of this novel inflammatory biomarker. This finding suggests that IL-40 may not only serve as an inflammatory mediator but also as an integrative biomarker reflecting the redox inflammation axis in sepsis.

In addition, antioxidant parameters (native thiol and total thiol) showed modest but significant negative associations with hemogram-derived indices (NLR, PLR, NLPR, and MPV), while disulfide levels correlated positively with these ratios. Although hematological indices capture systemic inflammation, their weaker correlations relative to thiol–disulfide markers suggest lower sensitivity in reflecting oxidative stress.

Collectively, these findings underscore that thiol–disulfide homeostasis is intricately linked with both classical (CRP, TNF-α) and emerging (IL-40) inflammatory markers, positioning it as a critical integrative biomarker at the intersection of oxidative and inflammatory cascades in sepsis. By combining mechanistic insight with clinical applicability, thiol–disulfide homeostasis, together with established and novel cytokines, provides a comprehensive biomarker framework for assessing disease severity and progression.

**Table 3. Correlation analysis between thiol-disulfide homeostasis parameters, inflammatory markers, and hemogram indices in the experimental groups.**

|  |  | Total thiol | Disulfide | NLR | PLR | NLPR | MPV | CRP | IL-40 | TNF-α |
|---|---|---|---|---|---|---|---|---|---|---|
| **Native thiol** | r | .985 | −.662 | −.335 | −.387 | −.369 | −.374 | **−.808** | **−.818** | **−.601** |
|  | p | <.001 | <.001 | .061 | .029 | .038 | .035 | **<.001\*** | **<.001\*** | **<.001\*** |
| **Total thiol** | r |  | -,521 | −.348 | −.363 | −.376 | −.333 | −.766 | **−.811** | **−.642** |
|  | p |  | ,002 | .051 | .041 | .034 | .063 | **<.001\*** | **<.001\*** | **<.001\*** |
| **Disulfide** | r |  |  | .144 | .336 | .190 | .397 | **.660** | **.518** | .182 |
|  | p |  |  | .433 | .060 | .299 | .025 | **<.001&** | **.002** | .320 |
| **NLR** | r |  |  |  | .265 | **.993** | .224 | .363 | .293 | .402 |
|  | p |  |  |  | .143 | **<.001†** | .219 | .041 | .104 | .022 |
| **PLR** | r |  |  |  |  | .291 | .245 | .163 | .151 | .160 |
|  | p |  |  |  |  | .107 | .176 | .372 | .411 | .383 |
| **NLPR** | r |  |  |  |  |  | .236 | .398 | .311 | .402 |
|  | p |  |  |  |  |  | .193 | .024 | .084 | .023 |
| **MPV** | r |  |  |  |  |  |  | .175 | .297 | .297 |
|  | p |  |  |  |  |  |  | .337 | .297 | .297 |
| **CRP** | r |  |  |  |  |  |  |  | **.887** | **.574** |
|  | p |  |  |  |  |  |  |  | **<.001\*** | **<.001\*** |
| **IL-40** | r |  |  |  |  |  |  |  |  | **.727** |
|  | p |  |  |  |  |  |  |  |  | **<.001\*** |

The analyses reveal that native thiol and total thiol levels show a negative correlation with CRP (∗: p<0.001), whereas disulfide levels exhibit a positive correlation with CRP (&: p<0.001). Additionally, the strong positive correlation between NLR and PLR (†: p<0,001) supports the association between inflammatory processes and thrombocytosis.

## Histopathological Findings

Histopathological examination of lung, liver, and kidney tissues was conducted using a validated semi-quantitative scoring system to evaluate inflammatory cell infiltration, vascular congestion, alveolar alterations, and parenchymal cellular degeneration. In the Control group (DMSO), no pathological alterations were observed; lung, liver, and kidney structures remained intact with histopathological scores consistently at zero, thereby confirming preserved tissue integrity (Table 4, Fig 1).

Histopathological analysis revealed that LPS-induced sepsis produced extensive multi-organ injury, with significant pathological alterations observed in the lung, liver, and kidney. Lung tissues demonstrated emphysematous changes, epithelial desquamation, diffuse vascular congestion, and interstitial inflammation. The liver displayed pronounced vascular congestion, hepatocyte vacuolar degeneration, and coagulative necrosis, while kidneys exhibited tubular necrosis, inflammatory infiltration, and diffuse congestion. These changes were reflected in markedly elevated histopathological scores across all evaluated parameters compared with controls ($p < 0.001$), confirming severe sepsis-induced structural damage.

In contrast, animals treated with Vitamin E prior to LPS challenge showed attenuated pathological severity. Pulmonary sections exhibited reduced emphysematous changes and less congestion, hepatic tissues demonstrated milder hepatocellular degeneration, and kidneys displayed decreased tubular necrosis and inflammatory cell infiltration. However, post-hoc statistical analysis indicated that these improvements did not consistently reach significance, with the exception of specific pulmonary parameters where Vitamin E exerted a significant protective effect ($p < 0.01$). Importantly, animals receiving Vitamin E alone displayed preserved histoarchitecture without pathological alterations, similar to controls.

Overall, these results demonstrate that LPS-induced sepsis causes profound structural damage to multiple organs, while Vitamin E confers partial histological protection, particularly in lung tissue. Although a single-dose regimen provided measurable benefit, the incomplete normalization of histopathological scores suggests that optimized dosing strategies may be required to achieve robust, multi-organ protection.

**Table 4. Histopathological Scoring of Sepsis-Induced Organ Damage and the Protective Role of Vitamin E: Lung, Liver, and Kidney Assessment.**

| Variable | DMSO (n=8) Mean±SD | E-Vit (n=8) Mean±SD | LPS (n=8) Mean±SD | LPS+Vit E (n=8) Mean±SD | p value |
|---|---|---|---|---|---|
| **Lung** | | | | | |
| Vascular congestion | 1,00±0,00[a] | 1,00±0,00[a] | 1,25±0,46[a] | 1,13±0,35[a] | 0,278 |
| Alveolar damage/ Emphysema | 0,00±0,00[a] | 0,00±0,00[a] | 2,25±0,71[b] | 2,13±0,83[b] | <0,001 |
| Bronchial epithelial damage | 0,00±0,00[a] | 0,00±0,00[a] | 2,25±0,71[b] | 2,00±0,53[b] | <0,001 |
| Interstitial inflammation | 0,00±0,00[a] | 0,00±0,00[a] | 1,38±0,52[b] | 1,38±0,52[b] | <0,001 |
| **Liver** | | | | | |
| Vascular congestion | 0,00±0,00[a] | 0,00±0,00[a] | 1,63±0,52[b] | 1,25±0,46[b] | <0,001 |
| Hepatocyte vacuolar degeneration | 1,00±0,00[a] | 1,00±0,00[a] | 2,00±0,53[b] | 1,88±0,35[b] | <0,001 |
| Coagulative necrosis | 0,00±0,00[a] | 0,00±0,00[a] | 2,00±0,93[b] | 1,88±0,83[b] | <0,001 |
| Portal area inflammation | 0,00±0,00[a] | 0,00±0,00[a] | 2,25±0,46[b] | 2,25±0,46[b] | <0,001 |
| **Kidney** | | | | | |
| Vascular congestion | 0,00±0,00[a] | 0,00±0,00[a] | 2,00±0,76[b] | 1,75±0,71[b] | <0,001 |
| Glomerular structural changes | 1,00±0,00[a] | 1,00±0,00[a] | 2,00±0,53[b] | 1,88±0,64[b] | <0,001 |
| Tubular necrosis | 0,00±0,00[a] | 0,00±0,00[a] | 1,88±0,83[b] | 1,63±0,74[b] | <0,001 |
| Inflammatory cell infiltration | 0,00±0,00[a] | 0,00±0,00[a] | 2,25±0,46[b] | 1,88±0,64[b] | <0,001 |

Values are presented as mean±standard deviation (SD). Histopathological scores were assessed using a semi-quantitative scale (0=none, 1=mild, 2=marked, 3=diffuse). Statistical comparisons among groups were performed using one-way ANOVA followed by Bonferroni-adjusted post-hoc tests. Groups sharing the same superscript letter do not differ significantly; different superscript letters indicate statistically significant differences ($p < 0.05$).

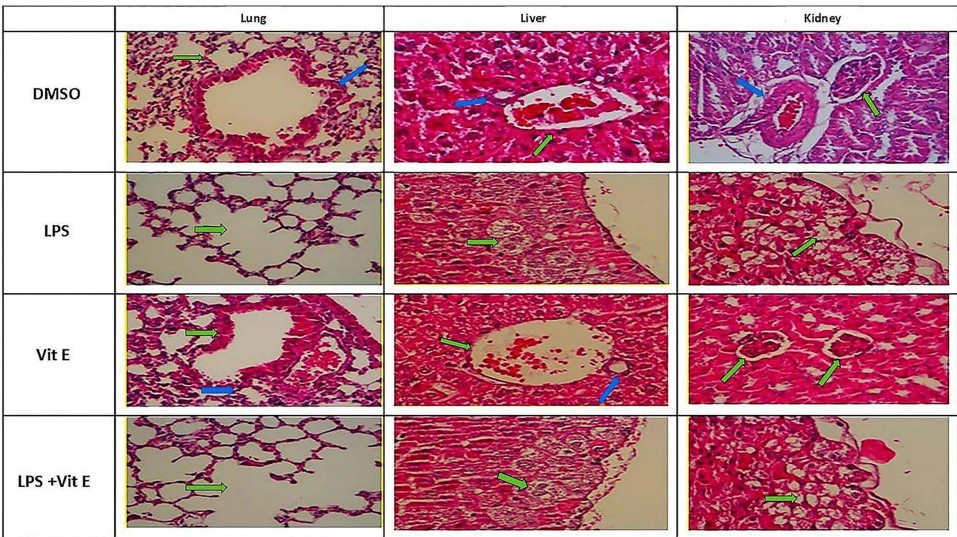

**Fig 1. Lung, liver and kidney tissue sections of the groups.**

## Discussion

This study evaluated the protective effects of vitamin E on oxidative stress, thiol–disulfide homeostasis, and inflammatory responses in an LPS-induced experimental sepsis model. Our findings demonstrate that vitamin E supplementation preserved thiol–disulfide homeostasis, reduced oxidative stress, and attenuated systemic inflammation, thereby mitigating tissue injury in septic mice. A particularly novel aspect of this work is the observation that Interleukin-40 (IL-40), a recently identified cytokine predominantly secreted by activated B cells and involved in both B-cell activation and regulation of NETosis [9] exhibited greater responsiveness to vitamin E treatment than TNF-α. This differential modulation highlights IL-40's potential as both an integrative biomarker of oxidative–inflammatory burden and a therapeutic target in sepsis. Consistent with recent reports showing that IL-40 inhibition prevents multi-organ injury and improves survival in sepsis models [9] our results suggest that vitamin E's protective effects extend beyond its established antioxidant actions to include selective modulation of emerging inflammatory pathways with possible clinical relevance.

Oxidative stress remains a central driver of sepsis-related organ dysfunction [5,6]. The observed reduction in native and total thiol levels with concomitant increases in disulfide levels in septic animals confirms the disruption of thiol–disulfide homeostasis. Restoration of these parameters in the LPS + Vitamin E group underscores vitamin E's capacity to stabilize redox balance, in line with literature emphasizing thiol–disulfide dynamics as sensitive indicators of oxidative stress in systemic inflammation [3,26]. Mechanistically, α-tocopherol protects cell membranes from lipid peroxidation, modulates protein kinase C, and suppresses NF-κB signaling, thereby reducing pro-inflammatory cytokine expression [11,12,19,20]. The decline in TNF-α observed in our model supports this mechanism, while enhanced antioxidant enzyme activities described in prior studies may further explain the preservation of thiol–disulfide homeostasis.

Systemic inflammation in sepsis involves multiple interacting pathways [2]. Clinical indices such as NLR, PLR, and CRP are widely applied prognostic tools [13–16,18,27]. In our study, vitamin E supplementation reduced CRP levels significantly, whereas improvements in NLR and PLR were modest, reflecting variable responsiveness of hematological markers to therapeutic interventions [28,29]. Importantly, vitamin E also suppressed IL-40 and TNF-α levels, reinforcing its dual antioxidant and anti-inflammatory actions. While TNF-α and IL-6 are established mediators of the septic cascade

[30–33] the stronger modulation of IL-40 in our study supports its potential role as a novel biomarker of disease activity and treatment response in sepsis [9].

Histopathological analyses corroborated the biochemical findings. LPS-induced sepsis resulted in severe pulmonary, hepatic, and renal damage, including vascular congestion, alveolar and tubular injury, and hepatocellular necrosis. Vitamin E conferred partial but consistent protection, with the most significant improvements observed in pulmonary structures ($p < 0.01$). Although protection in liver and kidney tissues was less pronounced, overall injury severity was reduced compared to untreated septic animals. Vitamin E alone caused no pathological alterations, supporting its safety profile. These results suggest that vitamin E can mitigate, but not completely prevent, sepsis-induced multi-organ injury.

## Limitations

This study has several limitations. First, only a single dose of vitamin E (100 mg/kg) was administered at one time point. Although this corresponds to a human equivalent of approximately 560 mg/day (well above typical dietary intake but within the therapeutic range [34] the effects of repeated or dose-escalated regimens remain unexplored. Determining the optimal dosing strategy and therapeutic window (e.g., preconditioning vs. therapeutic intervention) will require further investigation [19,35,36]. Second, measurement of IL-40, a cytokine highlighted in this study, is not yet routinely available in clinical laboratories, which currently limits its translational applicability. Nonetheless, our findings suggest that IL-40 holds promise as both a novel biomarker and a potential therapeutic target in sepsis. Finally, while the LPS-induced mouse model provides a reliable platform for studying acute systemic inflammation, it does not fully replicate the complexity and heterogeneity of human sepsis. Therefore, extrapolation to clinical practice should be made with caution. Carefully designed clinical studies remain essential to validate both the efficacy and safety of vitamin E as an adjunctive therapy in sepsis patients [37].

## Conclusion

In conclusion, vitamin E supplementation attenuated oxidative stress, preserved thiol–disulfide homeostasis, and reduced systemic inflammation in an LPS-induced sepsis model. Notably, modulation of IL-40 a recently described cytokine involved in B-cell activation and NETosis was more pronounced than that of TNF-α, highlighting its potential as a novel biomarker and therapeutic target. Histopathological findings further confirmed partial organ protection, most evident in pulmonary tissues. These results suggest that vitamin E holds promise as an adjunctive therapy in sepsis, though clinical translation will require optimized dosing strategies and validation in human trials.

## Supporting information

**S1 File. The minimal dataset necessary to replicate the study findings has been uploaded as Supporting Information (Minimal dataset thiol disulfide sepsis.xlsx).**
(XLSX)

## Acknowledgments

We gratefully acknowledge the guidance and support of Prof. Dr. Mehmet Emin Güldür, whose academic expertise and encouragement greatly contributed to the successful completion of this study.

## Author contributions

**Conceptualization:** Veli Fahri Pehlivan, Başak Pehlivan, Erdogan Duran, Abdullah Taskın, Yusuf Çakmak.

**Data curation:** Veli Fahri Pehlivan, Başak Pehlivan, Erdogan Duran, Abdullah Taskın, Ismail Koyuncu, Yusuf Çakmak.

**Formal analysis:** Veli Fahri Pehlivan.

**Funding acquisition:** Veli Fahri Pehlivan, Başak Pehlivan.

**Investigation:** Veli Fahri Pehlivan, Başak Pehlivan, Erdogan Duran, Abdullah Taskın, Ismail Koyuncu, Yusuf Çakmak.

**Methodology:** Veli Fahri Pehlivan, Abdullah Taskın, Ismail Koyuncu, Yusuf Çakmak.

**Project administration:** Veli Fahri Pehlivan.

**Resources:** Veli Fahri Pehlivan, Başak Pehlivan, Erdogan Duran.

**Software:** Veli Fahri Pehlivan, Başak Pehlivan, Erdogan Duran.

**Supervision:** Veli Fahri Pehlivan.

**Validation:** Veli Fahri Pehlivan, Ismail Koyuncu.

**Visualization:** Veli Fahri Pehlivan.

**Writing – original draft:** Veli Fahri Pehlivan, Başak Pehlivan.

**Writing – review & editing:** Veli Fahri Pehlivan, Başak Pehlivan, Erdogan Duran, Abdullah Taskın, Ismail Koyuncu, Yusuf Çakmak.

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
