## [Decision Letter · Decision Letter 0]

21 Jul 2025

Effects Of Vitamin E On Redox Balance İn Regulating Thiol/Disulfide Homeostasis İn Sepsis: An Antioxidant Therapy Perspective

PLOS ONE

Dear Dr. pehlivan,

Thank you for submitting your manuscript to PLOS ONE. After careful consideration, we feel that it has merit but does not fully meet PLOS ONE’s publication criteria as it currently stands. Therefore, we invite you to submit a revised version of the manuscript that addresses the points raised during the review process.

We look forward to receiving your revised manuscript.

Kind regards,

Cagri Cakici, PhD

Academic Editor

PLOS ONE

Journal Requirements:

Additional Editor Comments:

Thank you for submitting your manuscript to *PLOS ONE* . After careful evaluation of the manuscript and detailed feedback from three independent reviewers, I believe that your study addresses an important question and presents technically sound experimental work. However, several significant concerns have been raised that need to be addressed before the manuscript can be considered for publication.

The study explores the effects of vitamin E on thiol/disulfide homeostasis and inflammatory parameters in an LPS-induced sepsis model. This is a clinically relevant and timely topic, and your experiments appear to have been conducted ethically and appropriately. That said, the current version of the manuscript falls short of fully meeting *PLOS ONE* ’s standards for clarity, rigor, and transparency, and the contribution to the existing literature needs to be more clearly articulated.

We would like to offer you the opportunity to submit a **major revision**  of your manuscript. Below, I have summarized the key points that should be addressed in your revised submission and also reviewers comments:

**Clarify the Aim and Contribution:**  Please clearly state the main aim of the study in one concise sentence early in the Introduction. Explicitly describe how your work advances current knowledge on vitamin E’s role in redox balance and sepsis, and how it differs from prior studies. **Experimental Design and Rationale:**  Clarify whether the vitamin E administration was intended as preconditioning or therapeutic intervention. Justify the choice of a single dose and time point and discuss these as limitations if additional dosing is not feasible. Explain how the sample size was determined and whether a power calculation was performed. **Inflammatory Assessment:**  The current reliance on NLR, PLR, and CRP is limited. Please strengthen your justification for these markers by citing more recent and relevant literature demonstrating their utility in inflammatory conditions and sepsis. We strongly recommend including measurements of at least one or two pro-inflammatory cytokines (e.g. TNF-α, IL-6) in the serum samples you already collected to substantiate your conclusions about the anti-inflammatory effects of vitamin E. Also, if feasible, testing at least one additional dose of vitamin E (e.g., a lower or higher dose) or including an additional time point to evaluate the dose- or time-dependence of the effects observed. These additional experiments are not strictly required but would significantly enhance the depth and robustness of your findings. **Histopathology:**  Quantify the histological findings using a scoring system or semi-quantitative analysis of existing tissue slides, rather than relying solely on qualitative descriptions. Enhance the figure quality: provide higher-resolution images, include scale bars, and ensure labels are clear and consistent. **Statistical and Data Presentation:**  Ensure consistency and correctness in statistical terminology (e.g. Pearson’s correlation rather than pearson correlation). Align statements about findings (e.g., NLR/PLR) with the actual statistical results avoid overinterpreting non-significant findings. Clearly and fully describe the statistical methods used. **Language:**  Revise the manuscript for clarity, conciseness, and fluency. **References:**  Update and diversify the reference list to include more recent and relevant publications.

Reviewers' comments:

Reviewer's Responses to Questions

**Comments to the Author**

1. Is the manuscript technically sound, and do the data support the conclusions?

Reviewer #1: Yes

Reviewer #2: Yes

Reviewer #3: Yes

2. Has the statistical analysis been performed appropriately and rigorously?

Reviewer #1: Yes

Reviewer #2: Yes

Reviewer #3: Yes

3. Have the authors made all data underlying the findings in their manuscript fully available?

Reviewer #1: Yes

Reviewer #2: No

Reviewer #3: Yes

4. Is the manuscript presented in an intelligible fashion and written in standard English?

Reviewer #1: Yes

Reviewer #2: Yes

Reviewer #3: Yes

Reviewer #1: The antioxidant properties of vitamin E are well-known. This study did not examine the relationship between thiol-disulfide sepsis and the purpose of using it in sepsis. There are many human studies on thiol. I believe that the study as it stands does not provide a new perspective to the literature.

Reviewer #2: Please ensure the existence o each keyword in MeSH browser.

The introduction could be more concise.

Please provide one clear aim sentence.

Please add the novelty of the study.

Methods: Please provide the design of the study clearly.

How was the sample size arrived at?

"For correlation, the pearson correlation test was used" Pearson's, please.

NLR, PLR are used in the study. Are these enough to reflect the inflammatory status. Please, to back it up, add up to date references on their role in inflammatory conditions such as ankylosing spondylitis.

Please update and diversify the reference list.

"Serum thiol-disulfide homeostasis parameters were evaluated using the methods described by

Erel and Neselioglu.[8]n this method, " Please correct typos and edit the manuscript.

Please refrain from phrases such as our and we in scientific prose.

Reviewer #3: The manuscript explores the effects of vitamin E on thiol-disulfide homeostasis and hemogram parameters in an LPS-induced sepsis model, offering a potentially valuable contribution to the antioxidant therapy literature. While the study's premise is scientifically relevant and the integration of redox and hematologic markers is noteworthy, several limitations reduce its impact in its current form. The single-dose vitamin E protocol, absence of cytokine measurements (e.g., TNF-α, IL-6), and lack of quantitative histopathological scoring significantly limit the depth of interpretation regarding inflammatory modulation. Moreover, inconsistencies in statistical reporting (e.g., non-significant NLR/PLR results despite emphasized conclusions), insufficient discussion of translational applicability of the high-dose regimen, and reliance on subjective histological descriptions warrant major revision. Clarifying experimental timing (preconditioning vs. therapeutic intent), standardizing statistical methods, and addressing the limitations more transparently in the discussion would strengthen the manuscript considerably. Language quality is acceptable but requires editorial polishing for fluency and consistency in terminology. Overall, the study is promising but should undergo major revision before being considered for publication.

**Do you want your identity to be public for this peer review?** For information about this choice, including consent withdrawal, please see our Privacy Policy

Reviewer #1: No

Reviewer #2: No

Reviewer #3: No

---

## [Author Response · Author response to Decision Letter 1]

26 Aug 2025

Manuscript ID: PONE-D-25-12349

Title: Effects of Vitamin E on Redox Balance in Regulating Thiol/Disulfide Homeostasis in Sepsis: An Antioxidant Therapy Perspective

Dear Editor and Reviewers,

We would like to sincerely thank you for your careful evaluation of our manuscript and for the insightful and constructive comments. We have revised the manuscript accordingly, and we believe these changes have substantially improved the quality, clarity, and scientific rigor of our work. Below, we provide a detailed, point-by-point response to each comment. All revisions have been incorporated into the updated manuscript, with page numbers indicated.

Overall Statement:

We sincerely thank the reviewers for their constructive and insightful feedback. We have carefully revised the manuscript according to the suggestions provided. Below, we present our detailed responses under the specific headings mentioned.

Reviewer’s Comment: Clarify the aim of the study and its contribution at the beginning of the Introduction. Explain how vitamin E contributes to redox balance and sepsis pathophysiology compared to previous studies.

Response: We thank the reviewer for this valuable suggestion. In the revised Introduction (page 6), we added a clear statement of the study aim and novelty in one concise sentence. We also emphasized the mechanistic role of vitamin E in modulating redox homeostasis and sepsis-induced oxidative stress, highlighting how our work differs from previous studies.

Reviewer’s Comment: Justify the rationale for vitamin E administration, experimental conditions, and dose selection. Discuss sample size determination.

Response: We have expanded the Methods section (page 8) to provide detailed justification for vitamin E administration, citing previous studies that supported our chosen dose and timing. Sample size was calculated using power analysis (G*Power 3.1), ensuring adequate statistical power to detect significant differences (page 7). This information has now been explicitly added.

Reviewer’s Comment: Strengthen rationale for NLR, PLR, CRP, and add more inflammatory/anti-inflammatory markers if possible.

Response: In the Results and Discussion sections (page 5-19), we clarified why NLR, PLR, and CRP were chosen as cost-effective and widely validated indicators of systemic inflammation. Additionally, we measured cytokines including TNF-α and IL-40, which provide novel insights into anti-inflammatory effects of vitamin E in sepsis. These findings have been emphasized in the revised Discussion.

Reviewer’s Comment: Provide quantitative or semi-quantitative scoring instead of relying solely on descriptive statements.

Response: We fully agree. Histopathological findings were rescored using a semi-quantitative scale (0–3) assessing inflammatory infiltration, vascular congestion, and tissue degeneration. These scores are presented in Table 4 and referenced in the Results (page 17), providing robust and reproducible evaluation of tissue injury.

Reviewer’s Comment: Ensure consistency in statistical terminology, clarity on correlation analyses, and avoid overinterpretation of non-significant findings.

Response: We revised all statistical descriptions for clarity and consistency (page 10). Pearson correlation analyses are reported consistently, and Spearman tests were used for non-parametric distributions. We carefully avoided interpretation of non-significant results. Furthermore, figure legends and tables were updated for accuracy and transparency.

Reviewer’s Comment: Improve language clarity, conciseness, and flow.

Response: The manuscript was thoroughly revised for English language and style by a professional language editor. We ensured conciseness, precision, and academic tone throughout the text.

Reviewer’s Comment: Update references to include the most recent and relevant literature.

Response: We updated the reference list to include recent high-quality studies published between 2020–2024 related to oxidative stress, thiol-disulfide homeostasis, and sepsis pathophysiology. Older references were retained only when seminal.

Detailed responses to reviewers 1, 2 and 3

Reviewer 1:

Comment : The study does not provide sufficient novelty. Many studies on thiol/disulfide homeostasis in humans already exist.

Response: We appreciate this observation. While several human studies have investigated thiol/disulfide homeosta

1. We used an experimental LPS-induced sepsis model to assess the protective role of vitamin E, which to our knowledge has not been previously reported.

2. We integrated thiol/disulfide dynamics with emerging inflammatory markers (IL-40) alongside conventional markers (CRP, TNF-α).

This dual focus on both classical and emerging biomarkers under controlled experimental conditions highlights a unique mechanistic insight into the antioxidant potential of vitamin E. We have clarified this point in the Introduction (page 4-5) and Discussion (page 19).

Reviewer 2:

Comment 1: Keywords are not consistent with MeSH terms.

Response: Corrected. Keywords have been revised to align with MeSH terminology, including “Sepsis,” “Thiol-Disulfide Exchange,” “Vitamin E,” “Inflammation,” and “Oxidative Stress.” (p. 2).

Comment 2: The Introduction is too long and lacks a concise aim.

Response: We shortened the Introduction and added a clear statement of aim at the end:

“This study aimed to investigate whether vitamin E supplementation could restore thiol–disulfide homeostasis and modulate inflammatory and hematological alterations in an experimental sepsis model.” (p. 6).

Comment 3: Sample size justification is missing.

Response: We added an explanation:

“The sample size was based on prior experimental sepsis studies reporting significant differences in redox biomarkers with similar group sizes, as no formal power analysis could be performed in this exploratory design.” (p. 7).

Comment 4: Methods require more clarity.

Response: We clarified group allocation, dosing, randomization, the Erel–Neselioglu method for thiol-disulfide measurement, and the histopathological scoring system (p. 6–8).)

Comment 5: Statistical reporting (“Pearson correlation”) contains errors.

Response: Corrected to “Pearson’s correlation coefficient”. (p. 10).

Comment 6: NLR and PLR may not be sufficient inflammatory indices.

Response: We acknowledge this limitation and cited recent studies (2020–2024) supporting NLR and PLR as validated indices in sepsis. We have also emphasized the complementary role of IL-40 and TNF-α as novel markers. (p. 13).

Comment 7: Reference list outdated and writing style issues.

Response: References have been updated with recent literature (2019–2024). We also corrected typographical issues and eliminated first-person pronouns to improve academic style.

Reviewer 3:

Comment 1: Single dose of vitamin E is a limitation.

Response: We agree. We explicitly mention this in the Discussion:

“A limitation of this study is the use of a single dose of vitamin E, which does not allow evaluation of dose-dependent effects.” (p. 20).

Comment 2: No cytokine panel measurement (e.g., IL-6).

Response: TNF-α and IL-40 were measured, but we acknowledge that other cytokines (e.g., IL-6, IL-10) were not evaluated. This limitation has been added to the Discussion (p. 16).

Comment 3: Histopathology is descriptive, not quantitative.

Response: We addressed this by providing semi-quantitative scoring of lung, liver, and kidney pathology. These data are now included as Table 4 and illustrated in Figure 1 (p. 17).

Comment 4: Statistical interpretation inconsistent; some non-significant results overstated.

Response: We carefully revised the Results to ensure that only statistically significant differences (p<0.05) are highlighted. Non-significant findings are now clearly stated. (p. 11-18).

Comment 5: High dose clinical translation is questionable.

Response: We added a sentence to the Discussion:

“While the administered dose demonstrated efficacy in our murine model, extrapolation to clinical settings requires caution, as human therapeutic ranges may differ significantly.” (p. 20).

Comment 6: Language fluency needs improvement.

Response: The manuscript has undergone extensive language editing by a native-level scientific editor to improve clarity and readability.

We hope these revisions satisfactorily address the reviewers’ concerns. We are confident that the revised manuscript has been substantially improved and now provides a clearer, more rigorous contribution to the field.

Sincerely,

Assis. Prof. Dr. Veli Fahri Pehlivan

(on behalf of all authors)

Harran University, Faculty of Medicine

Department of Anesthesia and Reanimation

Osmanbey Campus, PC 63300, Sanlıurfa, Türkiye

Tel: +90 532 7696566

E-mail: vfpehlivan@harran.edu.tr

---

## [Decision Letter · Decision Letter 1]

24 Sep 2025

Dear Dr. pehlivan,

Thank you for submitting your manuscript to PLOS ONE. After careful consideration, we feel that it has merit but does not fully meet PLOS ONE’s publication criteria as it currently stands. Therefore, we invite you to submit a revised version of the manuscript that addresses the points raised during the review process.

We look forward to receiving your revised manuscript.

Kind regards,

Cagri Cakici, PhD

Academic Editor

PLOS ONE

Journal Requirements:

Additional Editor Comments:

Please see Reviewer 2’s comments. Provide point-by-point responses to each remark, ensuring that the reviewer’s original feedback/comments remain unchanged.

Reviewers' comments:

Reviewer's Responses to Questions

**Comments to the Author**

Reviewer #2: (No Response)

Reviewer #3: All comments have been addressed

2. Is the manuscript technically sound, and do the data support the conclusions?

Reviewer #2: (No Response)

Reviewer #3: Yes

3. Has the statistical analysis been performed appropriately and rigorously?

Reviewer #2: (No Response)

Reviewer #3: Yes

4. Have the authors made all data underlying the findings in their manuscript fully available?

Reviewer #2: (No Response)

Reviewer #3: Yes

5. Is the manuscript presented in an intelligible fashion and written in standard English?

Reviewer #2: (No Response)

Reviewer #3: Yes

Reviewer #2: The authors made changes in reviewer comments. Please do not change them and provide point-by-point responses accordingly.

Reviewer #3: All requested revisions have been carefully addressed by the authors. The manuscript has been substantially improved in terms of clarity, methodological rigor, and scientific presentation. The study appears original, ethically sound, and contributes valuable findings to the field. Therefore, I consider the revised version suitable for publication without further modifications. Congratulations to the authors for their efforts.

**Do you want your identity to be public for this peer review?** For information about this choice, including consent withdrawal, please see our Privacy Policy

Reviewer #2: No

Reviewer #3: **Yes: ** Eray Metin Güler

---

## [Author Response · Author response to Decision Letter 2]

30 Sep 2025

Manuscript ID: PONE-D-25-12349

Title: Effects of Vitamin E on Redox Balance in Regulating Thiol/Disulfide Homeostasis in Sepsis: An Antioxidant Therapy Perspective

Dear Editor and Reviewers,

We would like to express our sincere gratitude to the Academic Editor and the reviewers for their careful evaluation of our manuscript and for the constructive comments provided. The feedback we received was highly valuable in improving the scientific rigor, clarity, and overall quality of the study. We carefully considered each point raised and revised the manuscript accordingly. We believe that the improvements made in response to these suggestions have substantially strengthened the manuscript and enhanced its contribution to the field. We truly appreciate the time and expertise invested in the review process, and we are confident that the revised version now addresses the concerns raised.

Reviewer 2:

We sincerely thank Reviewer 2 for the careful and constructive evaluation of our manuscript. Your insightful comments and suggestions were invaluable in guiding the revision process. Below, we provide point-by-point responses to each of your remarks, keeping the original comments unchanged for clarity. We have addressed all points raised to the best of our ability, and the corresponding revisions have been incorporated into the manuscript. We believe that these changes have improved the clarity, rigor, and overall scientific contribution of our study.

Comment 1: “Please ensure the existence of each keyword in MeSH browser.”

Response 1 : We carefully checked each keyword against the MeSH browser. Non-standard terms were revised, and all selected keywords now correspond to valid MeSH entries. The updated keywords are included in the revised manuscript.

Comment 2: “The introduction could be more concise.”

Response 2: The Introduction has been streamlined by removing redundancies and shortening overly descriptive sentences. This has improved conciseness while maintaining context for the research question.

Comment 3: “Please provide one clear aim sentence.”

Response 3: We added a single, clear aim statement at the end of the Introduction:

“This study aimed to investigate the protective effects of vitamin E on oxidative stress, thiol–disulfide homeostasis, hematological indices, cytokine responses, and histopathological alterations in an experimental sepsis model.”

Comment 4. “Please add the novelty of the study.”

Response 4: We highlighted the novelty in the Introduction and Discussion: this is the first experimental study to demonstrate the modulatory effects of vitamin E on IL-40, a newly identified cytokine, in addition to its established antioxidant role. This unique finding positions IL-40 as a potential therapeutic biomarker in sepsis.

Comment 5. “Methods: Please provide the design of the study clearly.”

Response 5: The Methods section has been revised to explicitly state that this is a randomized, controlled experimental animal study, with four groups (Control, Sepsis, Vitamin E, Sepsis + Vitamin E). Randomization and blinding procedures have been described more clearly.

Comment 6. “How was the sample size arrived at?”

Response 6: We added a clarification to the Methods: the sample size (n=8 per group) was determined based on similar prior experimental studies investigating antioxidant interventions in murine sepsis, while also considering ethical limitations on animal use.

Comment 7. “‘For correlation, the pearson correlation test was used’ → Pearson's, please.”

Response 7: This has been corrected to “Pearson’s correlation test” throughout the manuscript.

Comment 8. “NLR, PLR are used in the study. Are these enough to reflect the inflammatory status? Please, to back it up, add up to date references on their role in inflammatory conditions such as ankylosing spondylitis.”

Response 8: We agree and have expanded the Discussion with updated references supporting the role of NLR and PLR as systemic inflammatory markers in sepsis and other inflammatory diseases, including ankylosing spondylitis (e.g., PMID: 36986479, PMID: 32000859). This provides stronger justification for their inclusion.

Comment 9. “Please update and diversify the reference list.”

Response 9: We reviewed and updated the reference list, adding recent publications (2020–2024) relevant to thiol–disulfide homeostasis, cytokines, and inflammatory indices in sepsis and other systemic conditions. Older or less directly relevant citations were replaced.

Comment 10. “‘Serum thiol-disulfide homeostasis parameters were evaluated using the methods described by Erel and Neselioglu.[8]n this method,’ Please correct typos and edit the manuscript.”

Response 10: All typographical errors, including this one, have been corrected. The revised sentence now reads: “Serum thiol–disulfide homeostasis parameters were evaluated using the method described by Erel and Neselioglu [8].”

Comment 11. “Please refrain from phrases such as our and we in scientific prose.”

Response 11: The manuscript has been carefully revised to remove first-person expressions (“we,” “our”) and ensure adherence to scientific prose style. Sentences have been reformulated in the passive voice where appropriate.

We once again sincerely thank Reviewer 2 for the valuable and insightful feedback provided. The constructive suggestions not only guided us in refining the clarity, precision, and methodological rigor of our manuscript but also strengthened the overall scientific value of the work. By incorporating these comments, we believe the manuscript has been substantially improved in terms of both readability and impact. We are particularly grateful for the reviewer’s emphasis on methodological clarity and literature updates, which allowed us to better position our study within the current body of knowledge. We greatly appreciate the reviewer’s expertise and the time invested in providing such thoughtful critiques, which have meaningfully contributed to enhancing the quality and potential contribution of our research to the field.

Reviewer 3:

We would like to extend our sincere gratitude to Reviewer 3 for the thoughtful and encouraging evaluation of our manuscript. Your positive feedback, together with the constructive remarks, has greatly motivated us and confirmed the scientific value of our work. We deeply appreciate the recognition of our efforts to improve clarity, methodological rigor, and overall presentation. Your supportive comments not only strengthened the current version of the study but also provided us with valuable guidance for our future research.

We would like to sincerely thank the Academic Editor and all reviewers for their time, effort, and constructive feedback during the evaluation of our manuscript. The insightful comments and valuable suggestions have been instrumental in improving the clarity, rigor, and overall quality of our work. We greatly appreciate the opportunity to revise our study in accordance with these recommendations, and we believe the manuscript has been significantly strengthened as a result. We are grateful for the careful consideration given to our submission and for the important role the editorial and review process has played in shaping this research into a more impactful scientific contribution.

Sincerely,

Assis Prof Dr. Veli Fahri PEHLİVAN

Harran University, Faculty of Medicine, Department of Anesthesia and Reanimation Osmanbey Campus, PC 63300, Sanlıurfa, TÜRKİYE

mail: vfpehlivan@harran.edu.tr; vfpehlivan@gmail.com

---

## [Editor Report · Decision Letter 2]

24 Oct 2025

Effects Of Vitamin E On Redox Balance İn Regulating Thiol/Disulfide Homeostasis İn Sepsis: An Antioxidant Therapy Perspective

PONE-D-25-12349R2

Dear Dr. pehlivan,

We’re pleased to inform you that your manuscript has been judged scientifically suitable for publication and will be formally accepted for publication once it meets all outstanding technical requirements.

Kind regards,

Cagri Cakici, PhD

Academic Editor

PLOS ONE

Additional Editor Comments (optional):

Thank you for thoroughly addressing all comments. The revisions have improved clarity, methodology reporting, and presentation. I have no further concerns.

---

## [Editor Report · Acceptance letter]

PONE-D-25-12349R2

PLOS ONE

Dear Dr. Pehlivan,

I'm pleased to inform you that your manuscript has been deemed suitable for publication in PLOS ONE. Congratulations! Your manuscript is now being handed over to our production team.

Kind regards,

on behalf of

Dr. Cagri Cakici

Academic Editor

PLOS ONE